# The Interplay Between Pulmonary Hypertension and Atrial Fibrillation: A Comprehensive Overview

**DOI:** 10.3390/cells14110839

**Published:** 2025-06-04

**Authors:** Danish Sultan, Bianca J. J. M. Brundel, Kondababu Kurakula

**Affiliations:** Department of Physiology, Amsterdam UMC, Vrije Universiteit, Amsterdam Cardiovascular Sciences, Heart Failure and Arrhythmias, Pulmonary Hypertension and Thrombosis, 1081 HZ Amsterdam, The Netherlands; h.m.d.sultan@amsterdamumc.nl (D.S.); b.brundel@amsterdamumc.nl (B.J.J.M.B.)

**Keywords:** pulmonary hypertension, atrial fibrillation, proteostasis, DNA damage, autophagy

## Abstract

Pulmonary hypertension (PH) is a progressive lung disease characterized by abnormal pulmonary vascular pressure and right ventricular (RV) dysfunction. Atrial arrhythmias, including atrial fibrillation (AF) and atrial flutter, are common in patients with PH and significantly contribute to disease progression and mortality. A bidirectional pathophysiological link exists between PH and AF, encompassing shared mechanisms such as endothelial dysfunction, DNA damage, autophagy, inflammation, and oxidative stress, as well as mutual risk factors, including diabetes, obesity, heart disease, and aging. Despite these shared pathways, limited research has been conducted to fully understand the intertwined relationship between PH and AF, hindering the development of effective treatments. In this review, we provide a comprehensive overview of the epidemiology of PH, the molecular mechanisms underlying the development of AF in PH, and the overlap in their pathophysiology. We also identify novel druggable targets and propose mechanism-based therapeutic approaches to treat this specific patient group. By shedding light on the molecular connection between PH and AF, this review aims to fuel the design and validation of innovative treatments to address this challenging comorbidity.

## 1. Introduction

Pulmonary hypertension (PH) is a complex and heterogeneous disease affecting the pulmonary vasculature, with an estimated prevalence of 1–3% in Western countries [1,2]. PH is clinically defined by an elevated mean pulmonary artery pressure (mPAP) greater than 20 mm Hg, increased pulmonary vascular resistance (PVR) due to vascular remodeling and obstruction, and dysfunction of the right ventricle (RV) dysfunction [1,2]. These hemodynamic and structural changes result in frequent episodes of decompensation in PH patients. Among PH subtypes, including pulmonary arterial hypertension (PAH) and chronic thromboembolic pulmonary hypertension (CTEPH), 10–33% of patients develop atrial arrhythmias (AA), such as atrial flutter (AFL) and atrial fibrillation (AF) (Table 1). Mechanistically, these arrhythmias are driven by electrical conduction abnormalities, right atrial stretch, fibrosis, and sympathetic overactivity [3]. The resulting loss of atrial contraction during the diastole reduces RV filling, decreases cardiac output, and precipitates clinical deterioration with features of heart failure. Interventions aimed at restoring sinus rhythm, including electrical cardioversion, antiarrhythmic medications, and catheter ablation strategies, have demonstrated potential to reverse cardiac decompensation and improve clinical outcomes. Notably, AA in PAH and CTEPH are associated with an increased risk of mortality, emphasizing the importance of achieving and maintaining sinus rhythm as a key therapeutic goal (Table 1) [4]. In PH patients, AF is considered as a severe clinical complication that exacerbates hemodynamic compromise, increases the risk of stroke, worsens symptoms, and complicates disease management. Restoring sinus rhythm in these patients is crucial, as the continuation of AF can worsen hemodynamic compromise and increase the risk of complications [5,6,7,8]. Despite the clinical significance of both PH and AF, research studies addressing both PH and AF in parallel remain limited. Given the significant impact of AF in PH patients, there is a pressing need to develop mechanism-based treatment strategies specifically tailored for this patient population. These strategies should address both the underlying PH and the associated AF, ultimately improving the outcomes and overall management of PH patients.

While the association between PH and AF has been recognized for several decades, predisposing risk factors, clinical manifestation, and corresponding molecular root causes for the development of AF in PH patients are unknown. Notably, atrial arrhythmias, including AF, are far more common in PH due to left heart disease (Group 2) and PH due to chronic lung diseases and hypoxia (Group 3) than in PAH (Group 1) [15,16]. A subset of Group 2 PH patients of heart failure with preserved ejection fraction (HFpEF), creating an additional substrate for AF development, making it difficult to determine cause or consequence [17,18]. Chronic elevation of left atrial pressure and volume overload in Group 2 PH promote atrial stretch, fibrosis, and electrical remodeling—factors known to drive AF [19].

Large-registry studies, such as the Swedish Pulmonary Arterial Hypertension Register (SPAHR), have demonstrated a relatively low AF prevalence in PAH, particularly in younger patients, challenging the generalized narrative of AF as a frequent complication in all PH subgroups [20]. The SPAHR analysis reported an AF prevalence of approximately 13% in the overall PAH population, with a markedly lower incidence in younger individuals [20].

Similarly, AF presents in various clinical forms, including paroxysmal (recurrent episodes of AF that resolve on their own within 7 days), persistent (last longer than 7 days or require intervention), and permanent AF (where sinus rhythm control can no longer be restored) [21]. These AF subtypes reflect different degrees of structural and electrical remodeling and may evolve over time in response to progressive cardiac stress [21,22]. In different PH cohort studies, the incidence and prevalence of subtypes of AF have been studied (Table 1), but to our knowledge, no clear relationship has been established yet. Furthermore, asymptomatic AF may go undiagnosed, yet still contribute to clinical deterioration and stroke risk [23]. Despite its importance, most studies do not stratify patients by AF subtype, making it difficult to determine whether molecular mechanisms differ between PH types.

Additionally, the longstanding pressure and volume overload in PH patients induce complex structural remodeling processes in the RV and right atrium (RA). However, while severe left atrial dilation is a well-established arrhythmogenic substrate for AF in left-sided heart diseases, including PH [24], the impact of RA dilation remains less definitive [25]. This paradigm raises critical questions about the relative contribution of right heart structural and electrophysiological changes to AF development in PH patients.

Moreover, little is known about the impact of AF on ventricular and lung function in PH. In patients with severe PH, the elevated pressure in the pulmonary arteries can impact the function of the right atrium. This may result in stretching and enlargement of the atrium, that may create a favorable environment for the onset of AF. Conversely, AF can impair atrial function and contribute to hemodynamic compromises, further worsening the condition of PH [4]. Despite the high clinical relevance, so far, little effort has been made to understand the molecular mechanism, how PH contributes to AF onset, and vice versa. These insights are important for developing patient-tailored treatment strategies. Interestingly, recent research findings disclose that several key molecular pathways, including proteostasis derailment by exhaustion of heat shock proteins, increase DNA damage, autophagy, endothelial dysfunction, inflammation, and oxidative stress underlie AF [21,26,27,28,29,30] as well as PH [31,32,33,34].

Finally, emerging evidence suggests that metabolic syndrome, such as obesity, insulin resistance, and dyslipidemia may contribute to both PH and AF development [35,36,37]. Adipose tissue, particularly epicardial fat, secretes pro-inflammatory cytokines and adipokines (e.g., activin A, p53, leptin, and caveolin-1) that can lead to endothelial dysfunction, fibrosis, and electrical remodeling in both atrial and pulmonary vascular tissues [35,36].

Over the past few decades, significant advancements have been made in both pharmaceutical and interventional therapies for PH and AF. Despite these advancements, there remains a critical need for the development of mechanism-based drugs to improve prognosis and reduce morbidity and mortality. This review discusses the epidemiology and prevalence of AF in PH, explores the molecular mechanisms underlying AF in the context of PH, identifies novel druggable targets, and discusses strategies for their validation. By elucidating the molecular interplay between PH and AF, this review aims to support the design and validation of innovative mechanism-based therapeutic approaches tailored to this specific patient population.

## 2. Clinical Classification of PH

PH is classified based on shared underlying disease mechanisms, clinical presentations, hemodynamic parameters, and therapeutic strategies. The latest guidelines from the European Society of Cardiology and the European Respiratory Society (ESC/ERS) categorize PH into five groups according to underlying causes and pathophysiological mechanisms (Figure 1). Given the strong clinical association between AF in specific subgroups of PH, namely pulmonary arterial hypertension (PAH, Group 1), PH associated with left heart disease (Group 2), and chronic thromboembolic pulmonary hypertension (CTEPH, Group 4), this review focuses particularly on these groups [4,5,15,38] (Table 1).

## 3. Epidemiology and Prevalence of AF in PH

PH is a significant global health concern which is impacted in every age group. As mentioned, AF is frequently observed in PH patients within Group 1 (pulmonary arterial hypertension, (PAH)), Group 2 (PH associated with left heart disease), and Group 4 (chronic thromboembolic pulmonary hypertension (CTEPH)) due to distinct yet interconnected mechanisms. In PAH, excessive pulmonary vasoconstriction, smooth muscle hypertrophy, and fibrosis lead to progressive right ventricular (RV) pressure overload and dysfunction, causing right atrial stretch, fibrosis, and electrical conduction abnormalities that predispose to AF. In Group 2 PH, elevated left atrial pressures from left ventricular dysfunction, valvular disease, or abnormal filling increase pulmonary venous pressure, resulting in atrial dilation and fibrosis, key drivers of AF. CTEPH involves chronic fibrotic clots and thromboemboli that obstruct pulmonary arteries, leading to increased pulmonary vascular resistance and RV strain, which also induce atrial remodeling. Tongers et al. were the first to report major implications of atrial arrhythmias in group 1 and group 4 PH patients. They reported a cumulative 11.7% incidence of supraventricular tachyarrhythmias with AF (5.62%) and AFL (6.49%) among the selected cohort (Table 1) [6]. In a group of patients with PAH, 22% of the patients were reported with incidence of AA [9]. A study using Johns Hopkins PH registry data of 317 patients with different types of PH revealed that 13.2% of patients developed AA [13]. In a 5-year follow-up study, Olsson et al. observed a cumulative prevalence of 25.1% AF and AFL in patients with PAH (*n* = 157) and CTEPH (*n* = 82) [8]. In a relatively big study cohort (*n* = 755), patients with different types of PH with elevated pre and post-capillary pressure showed higher prevalence of supraventricular tachycardia (29%) with AF as the most prevalent type of arrhythmia (21%) (Table 1) [14]. Shared mechanisms across these groups include chronic atrial stretch, fibrosis, inflammation, and hemodynamic instability, all of which promote AF development. The presence of AF further exacerbates hemodynamic compromise, impairing ventricular filling, worsening symptoms, and increasing the risk of adverse outcomes, underscoring the clinical importance of managing AF in these PH subgroups.

## 4. Molecular Mechanisms Underlying AF Development in PH

Several hemodynamic parameters change during PH development. The elevated mean pulmonary artery pressure and increased right atrium pressure in PH can lead to frequent decompensation, atrial remodeling, electrical conduction abnormalities, and structural changes in the atrial tissue [39]. The increased stress on the atria can result in atrial enlargement, fibrosis, and conduction alterations, which create a substrate for the re-entry and onset and maintenance of AF [40]. Furthermore, in PH, there is often activation of neurohumoral systems, such as the sympathetic nervous system and the renin–angiotensin–aldosterone system. These neurohumoral factors can promote inflammation and oxidative stress, which are known to contribute to AF development [13,41] (Figure 2).

The molecular mechanism of AF development in PH is multiplex and seems to share many common signaling pathways driven by multiple triggers. Emerging research identifies an overlap in various molecular mechanisms that are triggered during the disease progression of AF in PH. These include proteostasis derailment via the exhaustion of heat shock proteins (HSPs), induction of DNA damage via mitochondria dysfunction and reactive oxygen species (ROS) production, increase in autophagic protein degradation, thromboembolism, endothelial dysfunction, and inflammation as depicted below (Figure 3).

Proteins are complex macromolecules made up of a few to thousands of amino acid units and play a crucial role in the proper functioning of cells, including cardiomyocytes. Proteostasis plays an important role in preserving the integrity of proteins, which is maintained by protein quality control (PQC) and its derailment triggers specific stress-responsive and protein misfolding pathways. PQC prevents toxic aggregate formation by ensuring the correct folding, misfolding, and breakdown of malfunctioning proteins and it largely relies on chaperones, especially HSPs, protein degradation pathways, such as the ubiquitin–proteasome system (UPS) and macroautophagy pathways.

### 4.1. Role of Heat Shock Proteins

HSPs play a crucial role in the pathophysiology of PH and AF, particularly in modulating cellular stress responses and maintaining proteostasis [42,43,44]. In PAH, several studies have highlighted the involvement of HSP90 in vascular remodeling, a key factor influencing disease progression and outcomes in PAH patients [45]. Excessive proliferation of pulmonary artery smooth muscle cells (PASMCs), a hallmark of PAH, is closely linked to elevated HSP90 levels observed in both plasma and pulmonary arteriole membranes of PAH patients and experimental models [45]. HSP90 regulates proteins critical to PAH progression, and its mitochondrial accumulation in PASMCs contributes to abnormal cell proliferation and vascular remodeling—central features of PAH [45,46].

In contrast, the role of HSP70 in PH differs significantly. HSP70 is integral to the removal of damaged or misfolded proteins via the unfolded protein response (UPS), promoting cellular homeostasis [47]. However, in chronic thromboembolic pulmonary hypertension (CTEPH) patients and experimental models, HSP70 expression is notably downregulated, suggesting a diminished capacity to manage proteotoxic stress in this subtype of PH [48,49].

In AF, small heat shock proteins, particularly HSP27, are highly expressed in cardiomyocytes under normal conditions, where they stabilize cytoskeletal structures and support contractile and electrophysiological functions [50,51]. Experimental AF model systems, including HL-1 and canine atrial cardiomyocytes, have demonstrated a protective role of HSP27 in preserving cellular integrity and mitigating arrhythmogenic changes. However, during advanced stages of persistent AF, HSP27 levels in atrial tissue are significantly depleted, contributing to electrical conduction abnormalities, impaired calcium handling, altered action potential duration, and structural defects in cardiomyocytes [50]. However, in the advanced stages of persistent AF, levels of HSP27 in human atrial tissue become depleted, which is associated with structural damage and progression of the arrhythmia [52,53]. Similarly, HSP70, along with HSP27, has been linked to reduced incidences of post-operative AF, underscoring its protective role in AF [54,55,56]. Notably, experimental studies indicate that increasing HSP expression, either through genetic manipulation or pharmacological interventions, can mitigate pathological changes in both PH and AF. In particular, HSP27 has been shown to play a critical role in maintaining cardiac health by stabilizing cytoskeletal structures and counteracting stress-induced damage. These findings highlight the therapeutic potential of targeting HSPs to address the underlying mechanisms driving PH and AF progression [50,51].

### 4.2. Oxidative Stress and DNA Damage in PH and AF

Oxidative stress, marked by elevated ROS, has been widely implicated in DNA damage in both PH and AF [27,57,58]. NADPH oxidases (NOXs), particularly NOX1, NOX2, and NOX4, are major sources of ROS in PH, driving endothelial dysfunction, PASMC proliferation, and vascular remodeling, all hallmarks of PAH progression [59,60,61]. NOX-derived ROS activate growth factors, such as transforming growth factor beta (TGF-β1) and vascular endothelial growth factor (VEGF), further exacerbating oxidative damage and contributing to the pathophysiology of PH [62,63,64,65]. NOX-driven oxidative stress (NOX1, NOX2, and NOX4) also plays a key role in Group 2 Group 3, and Group 4 PH, contributing to endothelial dysfunction, myocardial fibrosis, and structural remodeling of the right ventricle [66,67]. The overexpression of NOX1 in AF results in increased Connexin 43 remodeling and subsequently microcirculatory dysfunction [68]. NOX2 overactivity triggers AF onset, leads to myofibril structural damage and abnormal sodium and potassium channel expressions, causing several electrical altercations [69,70]. Similarly, in human AF and in HL-1 derived cardiomyocyte model, NOX4 is associated with atrial fibrosis and mitochondrial dysfunction, leading to increased oxidative stress, abnormal calcium handling, and structural heart changes, all of which heighten susceptibility to AF [69,71,72]. This elevated oxidative stress led by the NOX supra family collectively shortens APD, promoting reentry and sustained AF [73,74].

Increased oxidative stress also activates DNA damage repair pathways, such as poly(ADP-ribose) polymerase 1 (PARP-1), in both PH and AF [26,33,75]. While PARP-1 facilitates DNA double-strand break repair, its overactivation depletes nicotinamide adenine dinucleotide (NAD^+^) stores, exacerbating mitochondrial dysfunction and inflammation [33]. Genetic studies in PH have identified mutations in BMPR2, SMAD9, and other DNA repair-related genes, which correlate with higher DNA damage and altered BMP signaling in PAH [33,76,77,78]. Studies on experimental models further suggest that inhibiting PARP-1 or other DNA repair proteins, such as PIM1, KU70, or EYA3, could provide therapeutic benefits by reducing cell proliferation and vascular remodeling [33,79,80,81].

Mitochondrial dysfunction is another key contributor to DNA damage in PAH and AF [82,83]. In PH, mitochondrial abnormalities, including reduced mitochondrial copy number and enhanced fission, correlate with increased glycolysis and ROS production [84,85,86]. PARP-1’s involvement in regulating mitochondrial energy metabolism suggests that it may contribute to these mitochondrial alterations [87]. In AF, mitochondrial DNA (mtDNA) damage, reflected in circulating markers such as 8-hydroxy-2′-deoxyguanosine (8-OHdG), escalates with disease progression [88]. PARP-1 overactivation in AF depletes NAD^+^, leading to impaired energy production and ROS-induced DNA damage, further worsening atrial remodeling and electrical dysfunction [89]. In addition, recent research findings revealed increased quantities of DNA lesions in atrial tissue and blood samples of patients with AF [90].

These findings highlight the central role of oxidative stress and DNA damage in driving structural and functional changes in PH and AF. Targeting ROS generation, DNA repair pathways, and mitochondrial dysfunction could present novel therapeutic avenues for both conditions.

### 4.3. Autophagic Protein Degradation

Autophagy, a highly conserved cellular process, plays a crucial role in cardiovascular diseases, particularly in PH and AF [31,42,91]. In various forms of PH, including idiopathic PAH (iPAH) and hypoxia-induced PH, autophagy markers such as light chain 3 (LC3)-B and its activated form LC3B-II are significantly elevated, along with decreased levels of p62, particularly in pulmonary arterial endothelial cells (PAECs) and PASMCs [92,93,94,95,96]. This suggests that autophagy is generally activated in response to PH, possibly contributing to disease progression by promoting vascular remodeling. The mechanistic target of rapamycin (mTOR) pathway plays a crucial role in regulating autophagy in PH, with pulmonary vascular remodeling associated with the downregulation of the mTOR pathway in lung tissues obtained from patients with PAH and in experimental PH animal models [93,97,98,99,100]. Furthermore, recent research has highlighted the role of other key players in autophagy regulation in PH, such as murine double minute 2 (MDM2), which is upregulated in both hypoxia-induced PH mouse models and kippah patients, leading to the degradation of angiotensin-converting enzyme 2 (ACE2), a key AMPK-dependent regulator in vasoconstriction [101]. In Group 1 PH patients, recent studies show that NLR-family pyrin domain containing 3 protein (NLRP3) activation exacerbates vascular remodeling, smooth muscle proliferation, and right ventricular fibrosis [102,103]. Additionally, hypoxia-inducible factor 1-alpha (HIF-1α) has been shown to induce pathological vascular remodeling in PAH by regulating autophagy-related genes [104,105].

In the context of AF, autophagy is involved in various aspects of disease progression, including atrial electrical, cellular, and energy metabolism remodeling [106]. Excessive autophagic protein degradation in AF is mainly regulated through endoplasmic reticulum (ER) stress, which is a critical factor in electrical and contractile dysfunction driving AF [106,107,108]. In a tachypaced mouse-derived HL-1 cardiomyocytes model, Wiersma et al. demonstrated that the overexpression of eukaryotic initiation factor 2α (eIF2α) prevents autophagy, which they also confirmed in a tachypaced *Drosophila* fly model by pharmacologically inhibiting ER stress with 4-phenyl butyrate (4PBA) [106]. Additionally, stress kinases, particularly JNK2, play a pivotal role in AF by orchestrating abnormal calcium handling [108]. JNK2 downregulates Cx43 and promotes diastolic calcium leak from the sarcoplasmic reticulum while simultaneously increasing the SR calcium content and creating a pro-arrhythmic environment [109,110] The AMPK pathway has also been implicated in AF [111], with the electrophysiological properties of atrial tissues in AMPK knockout mice showing an increase in atrial ectopic activity before the onset of AF, followed by delayed left atrial chamber enlargement [112,113]. Moreover, recent studies have identified a connection between AF and the mTOR signaling pathway [111,114], although further research is needed to understand the exact mechanism of mTOR driving autophagy in AF. The NLRP3 inflammasome has been identified as an important biomarker in many cardiovascular diseases, including AF [115,116], with the crosstalk between the PI3K/AKT/mTOR pathway and the NLRP3 inflammasome potentially influencing the autophagy response in AF [26]. HIF-1α has also been studied in AF in response to hypoxia in cardiomyocytes and in AF patients [117,118]. HIF-1α regulates VEGF which triggers AMPK-mediated transient autophagy [119]. Gramley et al. implicated a sustained increase in angiogenesis-related proteins, like HIF-1α and VEGF, in AF patients compared to SR patients, which suggests a direct role of HIF-1 in the regulation of autophagy in AF [120]. The intricate interplay between autophagy, mTOR/AMPK pathways, HIF-1α, and the NLRP3 inflammasome in PH and AF presents promising avenues for therapeutic interventions, with future research focusing on identifying specific autophagy markers, developing targeted therapies, and elucidating the complex interactions between autophagy, inflammation, and oxidative stress in cardiovascular diseases [115,117]. As research progresses, modulating autophagy may emerge as a promising strategy for managing PH and AF, ultimately improving patient outcomes in these challenging cardiovascular diseases.

### 4.4. Thromboembolism and Endothelial Dysfunction

Thromboembolism emerges as a significant complication in PH, particularly as the condition transitions from an acute to a chronic phase, characterized by endothelial dysfunction, platelet activation, inflammation, and coagulation abnormalities [121,122]. Numerous prospective studies on PAH have revealed enhanced regulation of adhesion molecules, including soluble tumor necrosis factor-like weak inducer of apoptosis (sTWEAK), P-selectin, vascular cell adhesion molecule-1 (VCAM-1), and platelet-derived microparticles (PMPs), which facilitate platelet adhesion and activation, strongly correlated with RV dysfunction and disease progression [123,124,125,126]. Endothelial dysfunction is also associated with a decrease in thrombomodulin expression, which plays a critical role in inhibiting thrombin generation and promoting anticoagulant pathways [126]. In the context of PAH, activated platelets contribute to thrombus formation in response to heightened stress within the pulmonary circulation, further exacerbated by prothrombotic factors, like thromboxane A2 and adenosine diphosphate [127,128,129]. Additionally, endothelial dysfunction can intensify the activation cascade of various coagulation factors, including fibrinogen, factor VIII, and von Willebrand factor, which together elevate thromboembolic risk, although their precise contributions to PAH progression remain debated [130,131]. The activation of the fibrinolytic system is crucial in regulating clot formation; however, an imbalance has been observed in PAH, with increased levels of plasminogen activator inhibitor-1 (PAI-1) linked to impaired fibrinolysis [132]. This phenomenon parallels findings in AF, where elevated P-selectin and VCAM-1 levels indicate platelet activation and an inflammatory state, respectively, alongside increased thromboxane A2 and fibrinogen levels that may enhance thrombosis risk in patients with AF and chronic kidney disease [133,134,135,136].

### 4.5. Inflammation

Inflammation is a prominent feature in the progression of both PH and AF [137]. Histopathological analyses of pulmonary vascular tissues from patients with PAH and corresponding animal models reveal diverse inflammatory infiltrates, including macrophages, lymphocytes, and mast cells, which contribute to the inflammatory cascade [137]. In a rat model, the presence of a genetic mutation in BMPR2 was found to intensify the inflammatory response, indicating that altered immunity may be a causative factor in PH rather than a mere consequence [138]. Tamosiuniene et al. investigated the role of TGF-β as a key mediator in the inflammatory response, demonstrating that its activation through a SMAD-dependent pathway triggers vascular remodeling and inflammation in T-cell-deficient rats [139]. Supporting these findings, Price et al. observed increased nuclear p65 and activation of the NF-κB signaling pathway in isolated PAECs from patients with iPAH, which is instrumental in the activation of pro-inflammatory cytokines, such as interleukin-6 (IL-6) and tumor necrosis factor-alpha (TNF-α) [140]. In response to endothelial dysfunction, various adhesion molecules (e.g., ICAM-1 and VCAM-1), chemokines (e.g., MCP-1), and pro-inflammatory cytokines (e.g., IL-1β) are upregulated, promoting leukocyte adhesion and infiltrating tissues, thereby exacerbating disease pathogenesis [141]. In AF, C-reactive protein (CRP), an acute-phase protein produced by the liver, has been implicated in the inflammatory process. Research by Dernellis et al. first demonstrated elevated CRP levels in AF patients, revealing a progressive increase from paroxysmal AF (ParAF) to persistent AF (PerAF), suggesting a correlation with disease risk and progression [142]. Subsequent studies have further clarified CRP’s role in inflammation and oxidative stress [143,144]. Additionally, the NLRP3 inflammasome has been implicated in AF-related inflammation, promoting the production of pro-inflammatory cytokines such as IL-1β and IL-18, ultimately leading to atrial remodeling and fibrosis [26,116]. Moreover, innate immunity plays a significant role in the inflammatory response in AF through Toll-like receptor (TLR) signaling. Studies in AF cohorts indicate that TLR signaling is upregulated, resulting in the production of pro-inflammatory and pro-fibrotic cytokines—including IL-6, IL-18, and IL-17A—which contribute to inflammation and subsequent structural and contractile remodeling processes in different stages of AF, such as ParAF and PerAF [145,146,147,148].

## 5. Novel Druggable Targets

Despite the significant clinical relevance, there have been no (pre)clinical trials addressing both PH and AF within a single model to date. In managing AF in patients with PH, rhythm control and rate control medications are commonly utilized [6,7,15]. Recent ESC/ERS guidelines [15] advocate for rhythm control agents, given their critical role in cardiac contractility under hemodynamic stress, despite substantial clinical challenges posed by both treatment strategies. While these guidelines emphasize tailored approaches, primarily utilizing class III antiarrhythmic drugs without negative inotropic effects (e.g., amiodarone), the advancement of the disease often renders these options unsuitable due to potential drug–drug interactions [149], underscoring the urgent need for developing more effective therapies for patients with concurrent PH and AF.

Several promising drug targets have emerged from preclinical research, aimed at restoring proteostasis, repairing DNA damage, mitigating mitochondrial dysfunction, and reversing inflammatory responses (Table 2). Notably, L-glutamine has been tested in clinical trials for both PH and AF, functioning as a potent inducer of HSP70, which helps to alleviate hemolysis and oxidative stress [133]. Additionally, targeting the proteostasis network can relieve ER stress; for instance, 4-phenylbutyrate (4PBA) has been recognized for its ability to decrease ER stress by reducing the burden of misfolded proteins, thereby preventing the activation of the UPR, critical in both AF and PH [106,150]. The inhibition of histone deacetylases (HDACs), particularly HDAC6, has also shown therapeutic promise by facilitating the degradation of misfolded proteins while mitigating PASMC proliferation in PH. Tubastatin A, an HDAC6-selective inhibitor, enhances microtubule stability and as such attenuates AF promotion and PAH [28,151]. Restoring NAD^+^ levels presents another strategy for reversing proteostasis disruption, as inhibiting PARP1 preserves NAD^+^ levels crucial for sirtuin function, with selective PARP1 inhibitors like ABT-888 showing potential benefits in both AF and PH [27,152]. Furthermore, nicotinamide, a precursor to NAD^+^, has demonstrated preventive effects on cardiomyocyte remodeling in AF models [27] and RV dysfunction in PH models [153]. Finally, while non-specific steroids and nonsteroidal anti-inflammatory drugs are frequently used to mitigate inflammation in both conditions, they often yield suboptimal patient responses [154,155,156], highlighting the need for targeted approaches, such as NLRP3 inflammasome inhibition. MCC950, a selective NLRP3 inhibitor, has been effective in reducing pro-inflammatory cytokine release in animal studies of both AF and PH, underscoring the importance of specifically targeting the NLRP3 pathway to address chronic inflammation [102,157,158].

## 6. Knowledge Gaps

Key knowledge gaps in PH and AF research include a lack of understanding of the molecular pathways that connect the two conditions, particularly specific signaling pathways and genetic factors. While several molecular pathways implicated in AF pathogenesis have been described in PH, these mechanisms are not unique to this condition and are shared across a spectrum of chronic cardiac and pulmonary diseases [162,163]. As such, the extent to which these pathways serve as specific contributors to AF in PH, rather than reflecting broader cardiovascular remodeling processes, requires further investigation. Additionally, there are insufficient data on the role of genetic predispositions and epigenetic modifications in their development and progression. Insights into the specific signaling and genetic pathways may fuel the identification of novel druggable mechanism-based targets [22]. The potential bidirectional relationship between PH and AF is also poorly understood, with unclear mechanisms regarding how each condition may exacerbate the other. These studies are challenging due to heterogeneous patient populations, multifactorial disease etiologies, and limited cardiovascular phenotyping, which hinder efforts to track PH development in patients with AF. Furthermore, research on the efficacy and tolerability of current treatment approaches, such as rate and rhythm control medications in PH patients with AF, is limited, and standardized protocols across studies are lacking, leading to variability in treatment strategies and outcomes. Moreover, there is a need for advanced in vivo and ex vivo models that accurately replicate the pathophysiological conditions of both PH and AF. As an example, the monocrotaline-induced PAH rat model (MCT) often develops RV dysfunction and can be used to study AF induction in PAH [164,165]. Moreover, longitudinal studies to track the progression of PH and AF together over time are also warranted. Finally, the impact of comorbidities commonly associated with PH and AF on clinical outcomes and treatment responses is not well understood, highlighting the need for further research in these areas to inform more effective management of these complex conditions.

## 7. Conclusions

The complex interplay between PH and AF represents a significant clinical challenge, necessitating a comprehensive understanding of their shared pathophysiological mechanisms. By illuminating the molecular intersections between these conditions, this review highlights the urgent need for multidisciplinary research efforts to develop targeted treatments. Future directions in PH-AF research should prioritize the identification of novel therapeutic targets and the validation of mechanism-based approaches through (pre)clinical trials. The integration of cutting-edge technologies, such as genomics and systems biology, will be essential for deciphering the intricate relationships between PH and AF. Moreover, the implementation of precision medicine principles will permit the development of tailored therapeutic strategies tailored to the unique characteristics of each patient. Ultimately, a deeper understanding of the PH-AF connection holds the promise of improving patient outcomes, quality of life, and reducing mortality, paving the way for a brighter future for patients affected by these devastating comorbidities.

## Figures and Tables

**Figure 1 cells-14-00839-f001:**
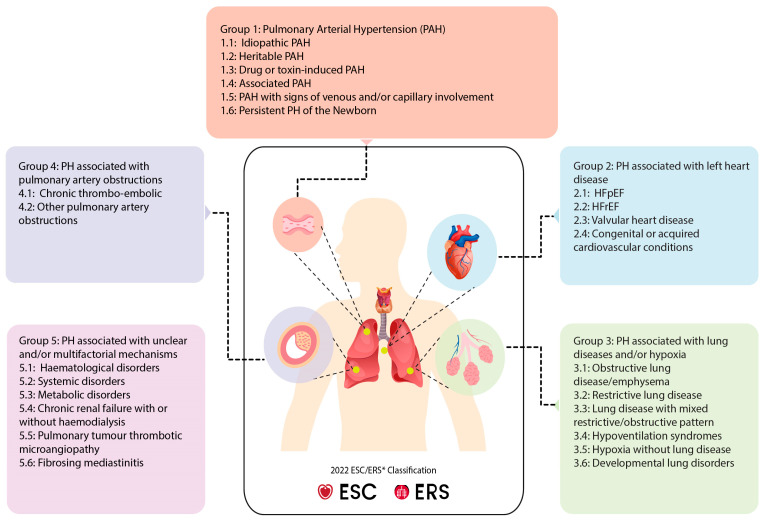
Clinical classification of PH according to 2022 ESC/ERS guidelines. PH is divided into five clinical groups that are further subdivided into subgroups based upon underlying disease mechanisms, clinical presentations, hemodynamic parameters, and therapeutic interventions. ESC: European Society of Cardiology, ERS: European Respiratory Society, HFrEF: heart failure with reduced ejection fraction, HFpEF: heart failure with preserved ejection fraction.

**Figure 2 cells-14-00839-f002:**
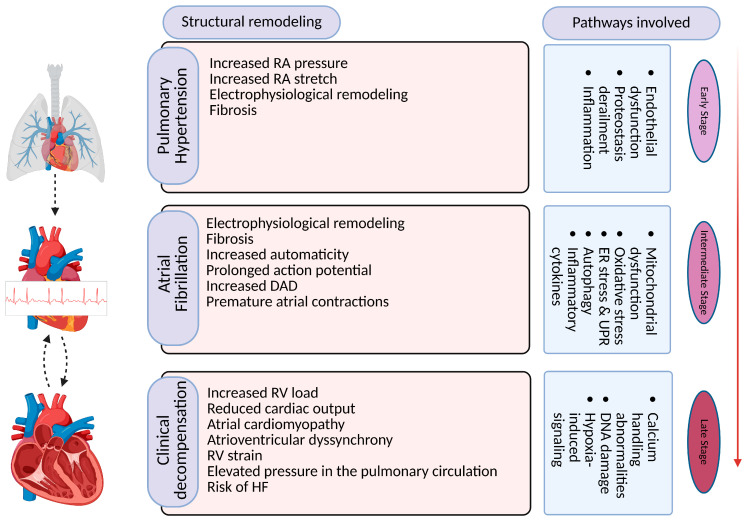
Overview of pathways and mechanisms involved in pathogenesis of AF in PH and associated pathophysiological implications. RA: right atrium, RV: right ventricle, DAD: delayed after depolarization, HF: heart failure. Created with BioRender.com (BioRender, RU28BW4XQD).

**Figure 3 cells-14-00839-f003:**
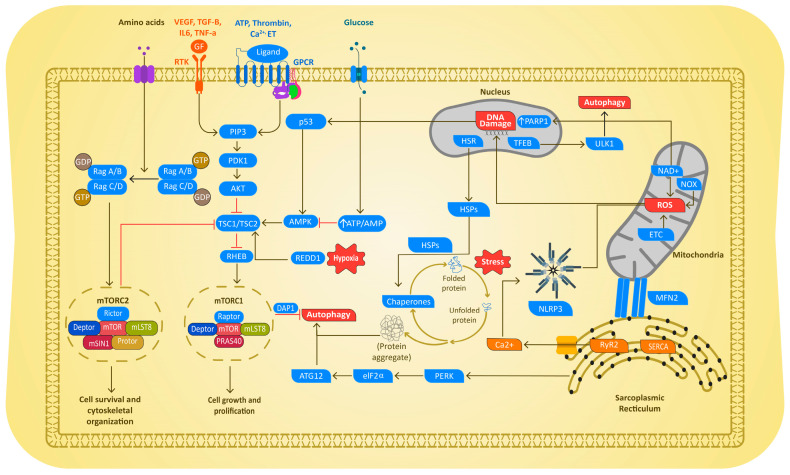
Schematic diagram depicting the molecular mechanisms driving AF in PH. In both cases, disease-mediated cellular stress caused proteostasis derailment, resulting in exhaustion of the cardioprotective HSP. The exhaustion of HSPs results in the activation of endoplasmic reticulum (ER) stress, resulting in excessive activation of the autophagic protein degradation pathway. PH and AF also cause mitochondrial dysfunction, ROS production, and subsequent DNA damage and Ca2+ abnormalities, as well as mTOR-induced autophagy followed by activation of inflammation and endothelial dysfunction (ED) pathways. RTK = receptor tyrosine kinases; GPCR = G protein-coupled receptors; NOX = NADPH oxidases GF = growth factors, PARP1 = poly(ADP-ribose) polymerase 1; mTOR 1/2 = Mammalian target of rapamycin 1/2; PDK1= serine/threonine kinase-3′-phosphoinositide-dependent kinase 1; TSC1/2 = tuberous sclerosis complex 1/2; AMPK = adenosine monophosphate-activated protein kinase; Redd1 = DNA damage response 1; Rheb: GTP binding protein Ras homolog enriched in brain; PDGF: platelet-derived growth factor; EGF: epidermal growth factor; VEGF: vascular endothelial growth factor; IL-6: interleukin 6; TNF-α: tumor necrosis factor alpha; TGF-β: transforming growth factor beta; ALK: anaplastic lymphoma kinase; ET: endothelin; ATP = adenosine triphosphate; CaSR: calcium-sensing receptor;.

**Table 1 cells-14-00839-t001:** Overview of studies showing prevalence of atrial arrhythmia in PH patients.

Registry	Sample Size	Patient Population *	Sex (%) *	Age (Years)	Incidence and Prevalence of Tachyarrhythmias	Ref.
Hannover Medical School, Hannover, Germany	231	iPAH = 70%, CTEPH = 12%	Female = 65%	48 ± 14 **	AFL = 6.49%, AF = 5.62%, AVNRT = 1.3%	[6]
PH Service of the Città della Salute e della Scienza of Turin.	77	PAH = 66%, CTEPH = 18%	Female = 53%	63(48–70.7) ***	ParAF = 3.8%, PerAF = 10.38%, LSPerAF = 3.8% AVNRT = 1.2%	[9]
Hannover Medical School, Hannover, Germany	239	PAH = 66%, CTEPH = 34%	Female = 61%	55 (49–66) ***	New onset of AF/AFL after 5 years = 25.1%	[8]
China	280	iPAH = 100%	Female = 68%	39 ± 15 **	AF = 5.71%, AFL = 4.64%, AT = 3.92%	[10]
University Hospital Bonn, Germany	64	PAH = 39%, CTEPH = 17%	Female = 47%	66.4 ± 11.8 **	AF = 32.1%	[11]
UPMC (Pittsburgh) hospitals	297	PAH = 90% CTEPH = 10%	Female = 67%	57.6 ± 14.7 **	AF = 15.48%, AFL = 8.41%	[12]
Johns Hopkins Pulmonary Hypertension Registry	317	iPAH = 37% SSc-PAH = 63%	Female = 84%	56.7 ± 14.4 **	AA (Total) = 13.2% AF = 5.99% AFL = 2.8%	[13]
General University Hospital, Prague	755	PAH = 44.23%, CTEPH = 17.21%	Female = 59%	60 ± 15 **	ParAF = 7.9% PerAF = 4.1% Permanent AF = 9% AF (total) = 21%	[14]

Round off values *, Mean **, Median ***, AA: atrial arrhythmia, AFL: atrial flutter, AT: atrial tachycardia, AVNRT: AV nodal reentry tachycardia, ParAF: paroxysmal atrial fibrillation, PerAF: persistent AF, LSPerAF: long-standing persistent AF, iPAH: idiopathic PAH, SSc-PAH: systemic sclerosis-associated PAH, CTEPH = chronic thromboembolic pulmonary hypertension.

**Table 2 cells-14-00839-t002:** Drugs with potential benefits that target key molecular modulators involved in PH and AF.

General Class	Drug(s)	Mechanism of Action	Condition	Trial Phase	Refs or Clinical Trial Identifier
**DNA damage**	ABT-888	PARP1 inhibitor	AF	Preclinical	[27]
			PH	Preclinical	[58,152]
	Nicotinamide	Enhances NAD^+^ and NADH levels, reducing oxidative damage to proteins and DNA	AF	Preclinical	[27,28]
			PH	Preclinical	[153]
**Oxidative stress**	L-glutamine	Induce HSPs and normalizes metabolites	AF	II	[159]
			PH	II	NCT01048905
**PQC system**	4-Phenyl-butyrate	ER stress inhibitor	AF	Preclinical	[106]
		ER stress signaling inhibitor	PH	Preclinical	[150]
	Tubastatin	HDAC6 inhibitor	AF	Preclinical	[28]
			PH	Preclinical	[151]
**Autophagy**	Chloroquine	Inhibition of autophagic flux [160]	AF	Preclinical	[161]
			PH	NA	NCT04314817
	MCC950	NLRP3-selective inhibitor	AF	Preclinical	[157]
			PF	Preclinical	[102]

AF: atrial fibrillation, ER: endoplasmic reticulum, HSP: heat shock protein, PH: pulmonary hypertension, PQC: protein quality control.

## Data Availability

Not applicable.

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
