# Peer review of "The Interplay Between Pulmonary Hypertension and Atrial Fibrillation: A Comprehensive Overview"

_cells, 2025, doi:10.3390/cells14110839_

Round 1

Reviewer 1 Report

Comments and Suggestions for Authors

In this article, Danis Sultan et al. review current knowledge on the association between pulmonary hypertension and atrial fibrillation. More specifically, they review molecular mechanisms described during PH and AF pathophysiology as evidence for interplay between the two diseases and as rational for common therapeutic targets. In my opinion there are two questions raise by PH and AF first how the two diseases aggravate each other and second, is there common underlying pathogenic mechanisms. These questions are insuffficientlty addressed 

  • One difficulty acknowledged by the authors, is the heterogeneity of PH. Clearly, PH secondary to heart failure (group 2) is a distinct entity in the context of AF. There is also a diversity of AF paroxysmal, persistent permanent or silent AF which are referring to distinct substrate. However, here this complexity is not really considered when addressing molecular mechanisms. May be a chapter dedicated to group 2 PH and AF could be proposed.
  • The co morbidities between PH and AF should also contribute to their interplay. For instance, metabolic disorders notably obesity should be discussed. In this line, molecular mechanisms associated with metabolic disorders such as the adipose tissue could be a potential common pathogenic factor in some clinical conditions associated with a high risk of AF and PH
  • The molecular mechanisms discussed by the authors are unspecific of the two diseases and can operate at different time course of the pathogenic process. May be a scheme indicating when these mechanisms could operate during the pathogenic process of PH and AF could be helpful.
  • Page 3 third paragraph, «…elevated pressure in pulmonary artery can impact the left atrium..” it should be the right?

Author Response

Reviewer 1

Comments and Suggestions for Authors

In this article, Danish Sultan et al. review current knowledge on the association between pulmonary hypertension and atrial fibrillation

Question 1:

In my opinion there are two questions raised by PH and AF: first, how the two diseases aggravate each other and second, is there a common underlying pathogenic mechanism. These questions are insufficiently addressed.

Answer 1:

We acknowledge the need to more clearly address how pulmonary hypertension (PH) and atrial fibrillation (AF) may aggravate each other, as well as whether common pathogenic mechanisms exist. However, research studies  addressing both PH and AF in parallel remain limited. To strengthen our manuscript in light of this, we have expanded the discussion by incorporating findings from available clinical studies that underscore the clinical impact of AF in patients with PH, by  contributing to hemodynamic deterioration, increased stroke risk, and worse clinical outcomes. Specifically, we have added the following sentence citing an original study to emphasize the clinical relevance (Line43);

“In PH patients, AF is considered  as a severe clinical complication that exacerbates hemodynamic compromise, increases the risk of stroke, worsens symptoms, and complicates disease management. Restoring sinus rhythm in these patients is crucial, as the continuation of AF can worsen hemodynamic compromise and increase the risk of complications [5-8]. Despite the clinical significance of both PH and AF, research studies  addressing both PH and AF in parallel remain limited.”

Furthermore, Table 1 provides a comprehensive overview of different cohort studies focusing the incidence of AF in various  PH cohorts. All studies suggest that restoring sinus rhythm in PH is beneficial. Additionally, to provide more details on the mechanistic insight of AF in PH we have created two updated  figures:

  • Figure 2 now provides an overview of the potential mechanisms involved in the pathogenesis of AF in the context of PH and outlines key pathophysiological consequences.

  • Figure 3 presents a detailed molecular overview of shared mechanisms driving both diseases, such as cellular stress-induced proteostasis derailment, HSP exhaustion, ER stress, autophagic pathway activation, mitochondrial dysfunction, ROS production, DNA damage, Ca²⁺ handling abnormalities, and mTOR-driven inflammation and endothelial dysfunction.

We believe these additions clarify the relationship between PH and AF, provide information on the potential pathophysiological mechanisms involved and as such more explicitly address the two questions raised.

Question 2:

One difficulty acknowledged by the authors, is the heterogeneity of PH. Clearly, PH secondary to heart failure (group 2) is a distinct entity in the context of AF. There is also a diversity of AF paroxysmal, persistent permanent or silent AF which are referring to distinct substrate. However, here this complexity is not really considered when addressing molecular mechanisms. May be a chapter dedicated to group 2 PH and AF could be proposed.

Answer 2:

 This is an interesting suggestion and while we appreciate the suggestion to include a separate section dedicated to group 2 PH, we respectfully note that this complex relationship has already been discussed in the context of left ventricular heart disease (see line 57). Given the broader scope of the current  review, which aims to highlight overarching mechanisms linking PH and AF across different subtypes, we believe that a dedicated chapter on group 2 PH would be  beyond the  scope of our paper. But to clarify this topic, we have expended the text by discussing  challenges due to group 2 PH in association with AF (Line 62);

“Subset of Group 2 PH patients of heart failure with preserved ejection fraction (HFpEF), creates an additional substrate for AF development, making it difficult to determine cause or consequence[19, 20]. Chronic elevation of left atrial pressure and volume overload in Group 2 PH promotes atrial stretch, fibrosis, and electrical remodeling—factors known to drive AF[21].”

We have also introduced additional clarifications regarding the different subtypes of AF. We adapted the text of the revised manuscript to adaptations in the text (line 74);

“Similarly, AF presents in various clinical forms, including paroxysmal (recurrent episodes of AF that resolve on their own within 7 days), persistent (last longer than 7 days or re-quire intervention), and permanent AF (where sinus rhythm control can no longer be re-stored)[23]. These AF subtypes reflect different degrees of structural and electrical remod-eling and may evolve over time in response to progressive cardiac stress[23, 24]. In differ-ent PH cohort studies, the incidence and prevalence of subtypes of AF have been studied (Table 1) but to our knowledge no clear relationship has been established yet. Furthermore, asymptomatic AF may go undiagnosed yet still contribute to clinical deterioration and stroke risk[25]. Despite its importance, most studies do not stratify patients by AF subtype, making it difficult to determine whether molecular mechanisms differ between PH types.”

Question 3:

The comorbidities between PH and AF should also contribute to their interplay. For instance, metabolic disorders notably obesity should be discussed. In this line, molecular mechanisms associated with metabolic disorders such as the adipose tissue could be a potential common pathogenic factor in some clinical conditions associated with a high risk of AF and PH

Answer 3:

Thank you for the suggestion. We agree with this comment and have expanded the manuscript to include a discussion on metabolic comorbidities and their potential contribution to both PH and AF (line 104);

“Finally emerging evidence suggests that metabolic syndrome such as obesity, insulin resistance, and dyslipidemia may contribute to both PH and AF development[37-39]. Adipose tissue, particularly epicardial fat, secretes pro-inflammatory cytokines and adipokines (e.g., Activin-1 p53, leptin, caveolin-1) that can lead to endothelial dysfunction, fibrosis, and electrical remodeling in both atrial and pulmonary vascular tissues[37, 38].”

Question 4:

The molecular mechanisms discussed by the authors are unspecific... A scheme indicating when these mechanisms could operate during the pathogenic process of PH and AF could be helpful.

Answer 4:

To improve the specificity of the mechanisms underlying PH and AF we have updated our schematic figure (Figure 2). We now provide information on all mechanisms that potentially are involved in the pathogenesis of AF in PH as well as the associated pathophysiological implications. The adapted figure distinguishes early, intermediate, and late-stage events.

Question 5:

Page 3 third paragraph, «…elevated pressure in pulmonary artery can impact the left atrium..” it should be the right?

Answer 5:

Thank you for pointing this out. It is indeed right atrium. We have now corrected this. (Line 93)

Reviewer 2 Report

Comments and Suggestions for Authors

This manuscript provided a comprehensive overview of the epidemiology of PH, the molecular mechanisms underlying the development of AF in PH, and the common ground in their pathophysiology. Overall, it is well written and easy to follow. However, authors might benefit from addressing the following points:

Major points:

  1. PH and AF have shared disease mechanism, and AF is a common complication of PH. Does a patient with AF often develop PH?
  2. What are procedures to prevent the development of AF in PH patients?

Minor points: None

Author Response

Reviewer 2:

Question 1:

PH and AF have shared disease mechanism, and AF is a common complication of PH. Does a patient with AF often develop PH?

Answer 1:

While PH is a well-documented contributor to the development of AF, to our knowledge there is no clinical data or observational studies available that provide conclusive information on patients with AF to develop . This point has been highlighted in our knowledge gap (line 464);

“The potential bidirectional relationship between PH and AF is also poorly understood, with unclear mechanisms regarding how each condition may exacerbate the other.”

We have further expanded the manuscript to elaborate this point Line (466);

“These studies are challenging due to heterogeneous patient populations, multifactorial disease etiologies, and limited cardiovascular phenotyping, which hinder efforts to track PH development in patients with AF.”

Question 2:

What are procedures to prevent the development of AF in PH patients?

Answer 2:

Preventing AF development in patients with PH relies primarily on optimizing the management strategies of PH. While no procedures exist that specifically target AF prevention in PH patients, several strategies may reduce atrial remodeling and electrical impairment, thereby lowering the risk of AF onset. Importantly, restoring sinus rhythm has been considered beneficial in observational studies. Also, recent ESC/ERS guidelines suggest using class III antiarrhythmic drugs without negative inotropic effects to attenuate AF (line 413);

“Despite the significant clinical relevance, there have been no (pre)clinical trials addressing both PH and AF within a single model to date. In managing AF in patients with PH, rhythm control and rate control medications are commonly utilized[6, 7, 17]. Recent ESC/ERS guidelines [17] advocate for rhythm control agents, given their critical role in cardiac contractility under hemodynamic stress, despite substantial clinical challenges posed by both treatment strategies. While these guidelines emphasize tailored approach-es, primarily utilizing class III antiarrhythmic drugs without negative inotropic effects (e.g., amiodarone), the advancement of the disease often renders these options unsuitable due to potential drug-drug interactions [152], underscoring the urgent need for developing more effective therapies for patients with concurrent PH and AF.”

Reviewer 3 Report

Comments and Suggestions for Authors

The authors present an interesting review exploring the interplay between pulmonary hypertension (PH) and atrial fibrillation (AF), with a particular focus on shared molecular mechanisms and potential therapeutic targets. The narrative is generally well-organized, and the inclusion of emerging pharmacological interventions adds translational value. However, several areas could benefit from additional depth and clarity. My specific comments are:

Major Comments

  1. While the authors correctly state that AF is prevalent in Group II and Group III PH (Line 60–61), the review does not sufficiently discuss the shared pathophysiological mechanisms that underlie this coexistence. It would strengthen the review to elaborate on mechanisms such as pulmonary venous hypertension, atrial remodeling, oxidative stress, and increased atrial pressure/stretch, which are highly relevant in these groups.

  1. The section titled “Autophagic protein degradation” effectively outlines ER stress and inflammasome involvement in AF, but stops short of drawing a strong mechanistic parallel with PH. Multiple studies (e.g., Chen et al., 2018; Sun et al., 2023; Al-Qazazi et al., 2022) demonstrate the role of ER stress and NLRP3 activation in the vascular remodeling and inflammation associated with PH, especially in right ventricular failure. A direct comparison summarizing these overlapping pathways would significantly add value.

  1. The authors briefly mention HIF-1α in the context of PH, but its role in AF—particularly in promoting atrial remodeling, fibrosis, and ion channel dysregulation—is underexplored. A dedicated subsection or expanded paragraph could enhance understanding of how hypoxia-driven signaling contributes to the electrophysiological and structural changes in both PH and AF.

  1. It is unclear which NADPH oxidase (NOX) isoforms the authors are referring to (Line 226–229). Please specify whether NOX2, NOX4, or other isoforms are implicated. Additionally, since both PH and AF are associated with oxidative stress and calcium handling abnormalities, a more comprehensive discussion on how NOX-mediated ROS production affects calcium homeostasis would be beneficial.

  1. The section on thrombosis and endothelial dysfunction would benefit from a more focused discussion on chronic thromboembolic pulmonary hypertension (CTEPH) and its mechanistic links to AF.

Minor Comments

  1. Please reorder the drugs in Table 2 to match their sequence of appearance in the main text for ease of cross-reference. MCC950, a selective NLRP3 inflammasome inhibitor, is discussed in the text but is missing from the table. Including it would provide a more complete list of targeted therapies.

  1. In line 257, suggest replacing to ‘hypoxia induced PH’.

Author Response

Reviewer 3:

Question 1:

While the authors correctly state that AF is prevalent in Group II and Group III PH (Line 60–61), the review does not sufficiently discuss the shared pathophysiological mechanisms that underlie this coexistence. It would strengthen the review to elaborate on mechanisms such as pulmonary venous hypertension, atrial remodeling, oxidative stress, and increased atrial pressure/stretch, which are highly relevant in these groups.

Answer 1:

Thank you for the suggestion. To address the potential underlying mechanism for AF in PH, we elaborate in the adapted manuscript in greater detail on this matter (line 257). In addition, we updated figure 2 to provide more detail on these mechanisms that drive pathogenesis of AF in PH and associated pathophysiological implications. Furthermore, in the revised manuscript we now elaborate on oxidative stress and DNA damage (line 257);

“NOX driven oxidative stress (NOX1, NOX2, NOX4), also play a key role in Group 2 Group 3 and Group 4 PH, contributing to endothelial dysfunction, myocardial fibrosis, and structural remodeling of the right ventricle[68, 69].”

Question 2:

The section titled “Autophagic protein degradation” effectively outlines ER stress and inflammasome involvement in AF, but stops short of drawing a strong mechanistic parallel with PH. Multiple studies (e.g., Chen et al., 2018; Sun et al., 2023; Al-Qazazi et al., 2022) demonstrate the role of ER stress and NLRP3 activation in the vascular remodeling and inflammation associated with PH, especially in right ventricular failure. A direct comparison summarizing these overlapping pathways would significantly add value.

Answer 2:

Thank you for the suggestion. We adapted the manuscript to elaborate in greater detail on autophagy in AF and  we now draw a parallel comparison with PH (line 316);

“In Group 1 PH patients, recent studies show that NLR family pyrin domain containing 3 protein (NLRP3) activation exacerbates vascular remodeling, smooth muscle proliferation, and right ventricular fibrosis[104, 105].”

(Line 326); “In a tachypaced mouse derived HL-1 cardiomyocytes model, Wiersma et al. demonstrate that overexpression of eukaryotic initiation factor 2α (eIF2α) prevents autophagy, which they also confirm in tachypaced Drosophila fly model by pharmacologically inhibiting ER stress with 4‐phenyl butyrate (4PBA)[110]. Additionally, stress kinases, particularly JNK2, play a pivotal role in AF by orchestrating abnormal calcium handling [111]. JNK2 down-regulates Cx43 and promotes diastolic calcium leak from the sarcoplasmic reticulum while simultaneously increasing SR calcium content and creating a pro-arrhythmic environment [112, 113]”

Question 3:

The authors briefly mention HIF-1α in the context of PH, but its role in AF—particularly in promoting atrial remodeling, fibrosis, and ion channel dysregulation—is underexplored. A dedicated subsection or expanded paragraph could enhance understanding of how hypoxia-driven signaling contributes to the electrophysiological and structural changes in both PH and AF.

Answer 3:

To provide detailed information on HIF1a in AF, we have expanded the discussion on HIF-1α to more comprehensively address its role in AF (line 342);

“HIF-1α has also been studied in AF in response to hypoxia in cardiomyocytes and in AF patients [120, 121]. HIF-1α regulates VEGF which triggers AMPK mediated transient autophagy [122]. Gramley et al. implicate a sustained increase in angiogenesis-related proteins like HIF-1α  and VEGF in AF patients compared to SR patients, which suggests a direct role of HIF-1 in regulation of autophagy in AF [123].”

Question 4:

It is unclear which NADPH oxidase (NOX) isoforms the authors are referring to (Line 226–229). Please specify whether NOX2, NOX4, or other isoforms are implicated. Additionally, since both PH and AF are associated with oxidative stress and calcium handling abnormalities, a more comprehensive discussion on how NOX-mediated ROS production affects calcium homeostasis would be beneficial.

Answer 4:

Thank you for the comment. We have now clarified in the revised manuscript that NOX1, NOX2 and NOX4 are the primary NADPH oxidase isoforms implicated in both PH and AF (line 260);

“The overexpression of NOX1 in AF results in increased Connexin 43 remodeling and sub-sequently microcirculatory dysfunction [68]. NOX2 overactivity triggers AF onset, leads to myofibril structural damage and abnormal sodium & potassium channel expressions, causing several electrical altercation [69, 70]. Similarly, in human AF and in HL-1 derived cardiomyocytes model, NOX4 is associated with atrial fibrosis and mitochondrial dysfunction, leading to increased oxidative stress, abnormal calcium handling, and structural heart changes, all of which heighten susceptibility to AF [69, 71, 72]. This elevated oxidative stress lead by NOX supra family collectively shorten APD, promoting reentry and sustained AF [73, 74].”

Question 5:

The section on thrombosis and endothelial dysfunction would benefit from a more focused discussion on chronic thromboembolic pulmonary hypertension (CTEPH) and its mechanistic links to AF.

Answer 5:

While we acknowledge the clinical and mechanistic relevance of CTEPH, a detailed discussion of this specific PH subgroup is beyond the scope of our current review which aims to provide a broader overview of shared molecular mechanisms between PH and AF. We have instead maintained a general focus on thrombosis and endothelial dysfunction as common contributors across PH subtypes, including but not limited to CTEPH (Line 352).

Minor Comments:

Question 1:

Please reorder the drugs in Table 2 to match their sequence of appearance in the main text for ease of cross-reference. MCC950, a selective NLRP3 inflammasome inhibitor, is discussed in the text but is missing from the table. Including it would provide a more complete list of targeted therapies.

Answer1:

Thank you for pointing this out. We adapted in our revised manuscript the table by mentioning the drug names in order of appearance in the text. (Table2).

Question 2:

In line 257, suggest replacing to ‘hypoxia induced PH’

Answer 2:

Thank you for pointing this out. The typo error has been fixed in our revised manuscript. 

Round 2

Reviewer 1 Report

Comments and Suggestions for Authors

In their answer 3 concerning the role of fat in the interplay between AF and pulmonary hypertension, the authors are refering to Activin-1 as a ^possibmle mediator. However, it is activinb A  PMID: 23525094

Author Response

Question 1:

In their answer 3 concerning the role of fat in the interplay between AF and pulmonary hypertension, the authors are refering to Activin-1 as a ^possibmle mediator. However, it is activinb A  PMID: 23525094

Answer 1:

Thank you for pointing this out. We agree with the correction and have revised the manuscript accordingly to refer to activin A in Line (105);

“Adipose tissue, particularly epicardial fat, secretes pro-inflammatory cytokines and adi-pokines (e.g., activin A, p53, leptin, caveolin-1) that can lead to endothelial dysfunction, fibrosis, and electrical remodeling in both atrial and pulmonary vascular tissues [37, 38].”

Reviewer 3 Report

Comments and Suggestions for Authors

The authors have answered all my questions in a satisfactory manner.

While I understand the authors' decision to concentrate on thromboembolism rather than drawing direct parallels with CTEPH, incorporating overlapping molecular targets could have strengthened the section on "thromboembolism and endothelial dysfunction."

This is a minor point and not essential.

Author Response

Question 1:

The authors have answered all my questions in a satisfactory manner.

While I understand the authors' decision to concentrate on thromboembolism rather than drawing direct parallels with CTEPH, incorporating overlapping molecular targets could have strengthened the section on "thromboembolism and endothelial dysfunction."

This is a minor point and not essential.

Answer 1:

We appreciate your understanding and thoughtful feedback. As noted previously, we have chosen not to draw direct parallels with CTEPH in order to maintain the focus of the review. However, we recognize the value of highlighting overlapping molecular targets, and we have ensured that the section on ‘Thromboembolism and endothelial dysfunction’ reflects mechanisms relevant across PH subtypes, including pathways that may also play a role in CTEPH. Thank you again for your helpful comments.